# Functional subregions of the human entorhinal cortex

Anne Maass[1,2*†], David Berron[1,2*†], Laura A Libby[3], Charan Ranganath[4†], Emrah Düzel[2,5,6,7†]

[1]Institute of Cognitive Neurology and Dementia Research, Otto-von-Guericke University Magdeburg, Magdeburg, Germany; [2]German Center for Neurodegenerative Diseases, Magdeburg, Germany; [3]Center for Neuroscience, University of California at Davis, Davis, United States; [4]Department of Psychology, University of California at Davis, Davis, United States; [5]Institute of Cognitive Neurology and Dementia Research, Otto-von-Guericke University Magdeburg, Magdeburg, Germany; [6]Institute of Cognitive Neuroscience, University College London, London, United Kingdom; [7]Center for Behavioral and Brain Sciences, Magdeburg, Germany

**Abstract** The entorhinal cortex (EC) is the primary site of interactions between the neocortex and hippocampus. Studies in rodents and nonhuman primates suggest that EC can be divided into subregions that connect differentially with perirhinal cortex (PRC) vs parahippocampal cortex (PHC) and with hippocampal subfields along the proximo-distal axis. Here, we used high-resolution functional magnetic resonance imaging at 7 Tesla to identify functional subdivisions of the human EC. In two independent datasets, PRC showed preferential intrinsic functional connectivity with anterior-lateral EC and PHC with posterior-medial EC. These EC subregions, in turn, exhibited differential connectivity with proximal and distal subiculum. In contrast, connectivity of PRC and PHC with subiculum followed not only a proximal-distal but also an anterior-posterior gradient. Our data provide the first evidence that the human EC can be divided into functional subdivisions whose functional connectivity closely parallels the known anatomical connectivity patterns of the rodent and nonhuman primate EC.

*For correspondence: anne.maass@med.ovgu.de (AM); david.berron@med.ovgu.de (DB)

†These authors contributed equally to this work

Competing interests: The authors declare that no competing interests exist.

## Introduction

The entorhinal cortex (EC) is a major hub within the medial temporal lobe that mediates hippocampal-neocortical communication (*Buzsáki, 1996*; *Lavenex and Amaral, 2000*). Together with the adjacent perirhinal cortex (PRC) and parahippocampal cortex (PHC), these brain regions form a neural circuitry that is critical for learning and memory (*Suzuki and Eichenbaum, 2000*; *Eichenbaum et al., 2007*). Virtually nothing is known about how hippocampal and neocortical connectivity with the EC is organized in humans. This lack of knowledge has significantly limited the development of neurobiological theories of memory and navigation and our understanding of the clinical impact of localized EC damage in the early stages of neurodegenerative conditions such as Alzheimer's disease (AD).

Neuroanatomical evidence from studies in rodents suggests that there are two parallel input pathways that convey spatial and non-spatial input into the hippocampus via the EC (*Knierim, 2006*; *van Strien et al., 2009*; *Witter et al., 2014*). Specifically, spatial information is conveyed from the postrhinal cortex (POR, thought to be homologous to the PHC in primates), which shows preferential connectivity with 'medial EC' (MEC). In contrast, non-spatial information is conveyed from the PRC to the 'lateral EC' (LEC). LEC and MEC, in turn, are differentially connected with hippocampal subfields (i.e., subiculum and CA1) along the proximo-distal (transverse) axis. Projections of the LEC preferentially target the region close to the border between CA1 and subiculum (distal CA1 and proximal subiculum), whereas the MEC

**eLife digest** In the early 1950s, an American named Henry Molaison underwent an experimental type of brain surgery to treat his severe epilepsy. The surgeon removed a region of the brain known as the temporal lobe from both sides of his brain. After the surgery, Molaison's epilepsy was greatly improved, but he was also left with a profound amnesia, unable to form new memories of recent events.

Subsequent experiments, including many with Molaison himself as a subject, have attempted to identify the roles of the various structures within the temporal lobes. The hippocampus—which is involved in memory and spatial navigation—has received the most attention, but in recent years a region called the entorhinal cortex has also come to the fore. Known as the gateway to the hippocampus, the entorhinal cortex relays sensory information from the outer cortex of the brain to the hippocampus.

In rats and mice the entorhinal cortex can be divided into two subregions that have distinct connections to other parts of the temporal lobe. The 'medial entorhinal cortex' is the subregion nearest the centre of the brain, and it predominantly connects to parahippocampal cortex, which is involved in processing visual scenes. The other subregion, the 'lateral entorhinal cortex', is to the left or right of the center and has particularly strong connections to the perirhinal cortex, which is involved in the memory of objects. The two subregions are also connected to different parts of the hippocampus.

For many years researchers had assumed that the connectivity of the human entorhinal cortex was quite similar to that observed in rats and mice. However, it was not possible to check this as the entorhinal cortex measures less than about 1 cm across, which placed it beyond the reach of most commonly available brain-imaging techniques. Now, two independent groups of researchers have used ultra high-resolution functional magnetic resonance imaging (fMRI) to reveal a more complex structure in humans. The fMRI data reveal that the entorhinal cortex is divided into an anterior-lateral (to the front and at the side) subregion and a posterior-medial (to the back and at the centre) subregion in humans.

One of the groups—Maass, Berron et al.—used the imaging data to show that the anterior-lateral and posterior-medial subregions of the entorhinal cortex form distinct patterns of connections with the perirhinal cortex and the parahippocampal cortex, as well as with different parts of the hippocampus. The other group—Navarro Schröder, Haak et al.—studied functional connections across the whole neocortex to come to the same conclusions.

The discovery of these networks in the temporal lobe in humans will help to bridge the gap between studies of memory in rodents and in humans. Given that the lateral entorhinal cortex is one of the first regions to be affected in Alzheimer's disease, identifying the specific properties and roles of these networks could also provide insights into disease mechanisms.

preferentially projects to proximal CA1 and distal subiculum (e.g., *Witter et al., 2000a*; *Henriksen et al., 2010*). These partially segregated pathways have been differentially associated with the processing of, and memory for, object and context information (e.g., *Hunsaker et al., 2013*; for review see; *Knierim et al., 2014*; *Ranganath and Ritchey, 2012*; *Ritchey et al., in press*). Moreover, a recent network analysis on the rat connectome highlighted the LEC as a major cortical hub, forming the richest set of association connections of any cerebral cortical region (*Bota et al., 2015*).

Notably, although terminology for EC subdivisions in the rat emphasizes the lateral to medial axis, these areas do not differ solely with respect to their position in relation to the hippocampal formation and the rhinal fissure (*Witter et al., 2000a*). In actuality, LEC occupies the rostrolateral portion of the EC, whereas MEC occupies the caudomedial portion of the EC. In primates, ventral hippocampus and the adjacent EC are situated in a relatively more rostral position in the anterior temporal lobe. Although the position of the EC on the cortical surface and the orientation of anatomical axes differ across species, the relative topography of EC connectivity seems to be preserved in nonhuman primates. Anatomical studies suggest that the PRC is predominantly interconnected with the anterior third of the EC, whereas the PHC is predominantly interconnected with approximately the posterior two-thirds (*Suzuki and Amaral, 1994*). In addition, PRC/PHC connectivity with EC differs between

lateral and medial domains (*Suzuki and Amaral, 1994*). A functional distinction between anterior and posterior EC has been substantiated by a single-unit recording study of grid-cell-like neurons in nonhuman primates (*Killian et al., 2012*).

To summarize, findings from rodents and nonhuman primates suggest that connectivity of the human EC might differ along the longitudinal and lateral-medial axis, but because anatomical tracing studies cannot be performed in humans, direct evidence for this idea is lacking. Recent structural and functional MRI studies have assumed a lateral-medial distinction in humans according to the rodent nomenclature of LEC and MEC. However, whether such a lateral-medial dissociation of EC connectivity exists in humans and/or whether connectivity differs along the anterior-posterior EC axis has not been assessed so far.

Numerous studies have demonstrated that networks of brain regions linked by direct and indirect anatomical connections exhibit temporally coherent, low-frequency fluctuations in blood oxygenation level dependent (BOLD) functional magnetic resonance imaging (fMRI) data during both the resting and task states. Recent work has demonstrated that spatially contiguous but anatomically distinct brain regions can be reliably differentiated based on functional connectivity profiles measured with BOLD fMRI (for a review see *Fox and Raichle, 2007*). Using fMRI at 3 Tesla, human resting-state fMRI studies (*Kahn et al., 2008*; *Libby et al., 2012*) have reported reliable differences in connectivity between the PRC and PHC with the hippocampus. Whereas the PRC showed higher functional connectivity with the anterior hippocampus, the PHC showed stronger connectivity with the posterior hippocampus, a dissociation that was most evident for subiculum and CA1 subfields (*Libby et al., 2012*). Due to limitations in signal-to-noise ratio (SNR) and spatial resolution, these studies were unable to assess functional connectivity with the EC.

Here, we used ultra-high resolution fMRI at 7 Tesla to characterize the functional organization of the human EC. The high SNR of MRI data collected at 7 Tesla makes it feasible to acquire BOLD fMRI data at an unprecedented level of anatomical detail (e.g., *Maass et al., 2014*). To determine the topographic organization of EC connectivity in humans, we conducted two experiments in which fMRI data were acquired with a resolution of 0.8 mm (isotropic). Notably, this level of spatial resolution is more than six times higher than in previous high-resolution fMRI studies that investigated intrinsic connectivity within the MTL (*Lacy and Stark, 2012*; *Libby et al., 2012*). In two independent samples, we examined correlations of activation between individually anatomically defined PRC vs PHC seeds and the EC along its anterior-posterior and lateral-medial axis. In addition, we tested whether functionally distinct EC subregions also exhibited differential connectivity with the subiculum along a transverse or longitudinal hippocampal axis. In addition, we also analyzed connectivity profiles of PRC and PHC seeds with the subiculum. The present study, which focuses on patterns of EC connectivity within the MTL is complemented by a study by *Navarro Schröder et al. (2015)*, which examined connectivity topography of the EC with extended cortical networks outside of the MTL.

## Results

### Entorhinal connectivity topography related to PRC and PHC seeds
#### Seed-to-voxel connectivity of PRC vs PHC with EC
We used 7T high-resolution fMRI to characterize and compare intrinsic functional connectivity profiles of PHC and PRC seed regions with the EC. Particularly, we investigated correlations of activity over time (spontaneous changes in the BOLD signal) between individually defined PRC and PHC seed regions and the EC across subjects. Seed-to-voxel correlation analyses were performed on the native (unnormalized) residual fMRI data after extraction of task-related activity (see 'Materials and methods' section for details). Resulting connectivity maps were normalized on a group-specific T1-template by means of Region of Interest-Advanced Normalization Tools (ROI-ANTS; *Klein et al., 2009*; *Yassa and Stark, 2009*; *Avants et al., 2011*). The template has the same resolution and alignment as the functional images, such that coronal images (as in *Figure 1*) are oriented orthogonal to the longitudinal axis of the hippocampus. Note that our terminology of axes (anterior-posterior, lateral-medial and dorsal-ventral) always refers to the longitudinal hippocampal axis, although the hippocampus has a slightly upward-tilted position when moving from anterior to posterior within the brain. Group results maps were masked with a manually-defined EC ROI.

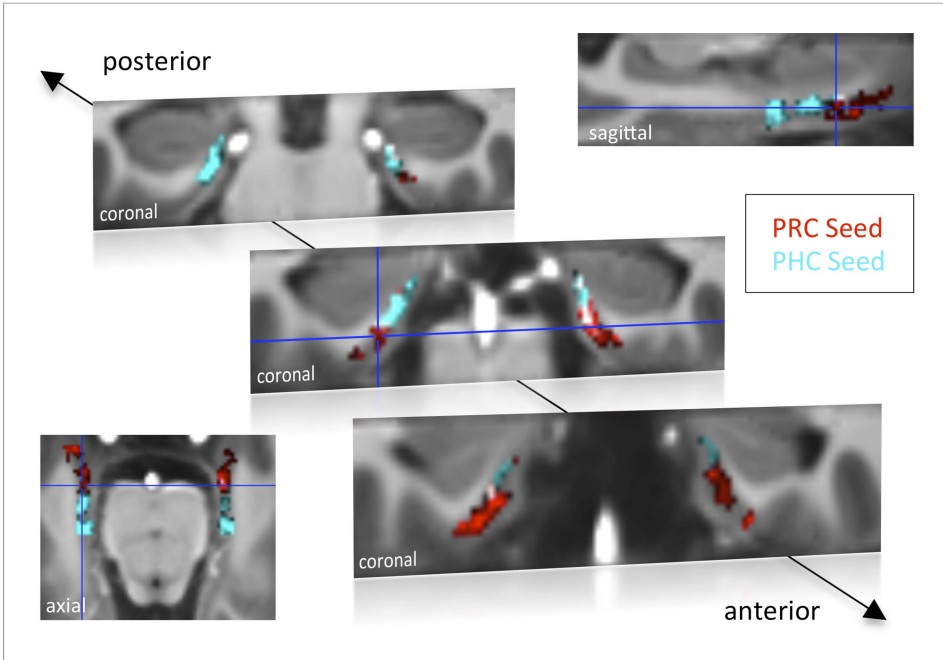

**Figure 1**. Functional connectivity profiles of parahippocampal cortex (PHC) and perirhinal cortex (PRC) seeds with the EC in Experiment 1. Group results for seed-to-voxel connectivity of bilateral PRC and PHC seeds with the EC shown for Experiment 1 (one-sample t-test; Z > 2.3, $p_{cluster}$ < 0.05, $N_{Exp.\ 1}$ = 15). Bright regions denote overlapping connectivity with PRC/PHC. Single-subject beta maps were normalized on the group-specific T1-template by ROI-based alignment with ANTS and masked with a manually defined EC ROI. The T1-template has the same resolution (and alignment) as the high-resolution functional EPI volumes (0.8 mm × 0.8 mm × 0.8 mm). See also *Figure 1—figure supplement 1* for results of Exp. 2. ROI: region of interest.
The following figure supplement is available for figure 1:

**Figure supplement 1**. Functional connectivity profiles of PHC and PRC seeds with the EC for Experiment 2.

Group-level functional connectivity profiles of each seed region for Experiment 1 and 2 are shown in *Figure 1* and *Figure 1—figure supplement 1*, respectively (voxelwise one-sample t-tests, Z > 2.3, $p_{cluster}$ < 0.05). While the PRC showed significant connectivity with bilateral EC clusters covering approximately the anterior two-thirds of the EC, significant functional connectivity of the PHC was found with bilateral EC clusters comprising about the posterior two-thirds of the EC (see also peak coordinates of significant clusters in *Table 1*). Additionally, PRC-connectivity clusters were limited to progressively more lateral regions of the EC when moving posteriorly while PHC-connectivity clusters were limited to progressively more medial regions of the EC when moving anteriorly. Overlapping connectivity with both seeds was strongest in the transition zone between anterior-posterior and lateral-medial EC (see bright regions in *Figure 1* and *Figure 1—figure supplement 1*).

Paired t-tests revealed those EC regions that exhibited significant stronger connectivity with PRC than PHC seeds, and vice versa (see *Table 1*). While stronger functional connectivity of the PRC was found with bilateral clusters in the anterior-lateral EC (al-EC), the PHC showed relatively stronger connectivity with bilateral clusters located in the posterior-medial EC (pm-EC). Paired t-test results are illustrated for Experiment 1 on *Figure 2*.

## Three-dimensional topography of entorhinal connectivity
In order to assess and visualize the 3-dimensional topography of differential EC connectivity with PRC vs PHC seeds, we plotted the connectivity preference of each voxel along the x-y-z direction (see *Figure 2B* and *Figure 2—figure supplement 1A*). Connectivity preference was defined on the basis of the paired t-test t-maps (red: $T_{PRC > PHC}$ > 0; blue: $T_{PHC > PRC}$ > 0). These plots indicated a complex 3-dimensional topography of EC connectivity with a gradient of PRC-to-PHC preference

**Table 1.** Univariate group-results for seed-to-voxel connectivity of PRC and PHC seeds with the EC

| | Cluster | Cluster | Cluster | Peak | Peak coordinate (template) | | | |
|---|---|---|---|---|---|---|---|---|
| | $P_{FWE-corr}$ | $P_{FDR-corr}$ | Size | Z-score | x | y | z | Side |
| Experiment 1 | | | | | | | | |
| PRC seed | <0.001 | <0.001 | 517 | 5.05 | 153 | 149 | 8 | R |
| | <0.001 | <0.001 | 273 | 4.82 | 108 | 146 | 12 | L |
| PHC seed | <0.001 | <0.001 | 380 | 5.29 | 150 | 129 | 9 | R |
| | <0.001 | <0.001 | 510 | 5.05 | 106 | 138 | 12 | L |
| PRC > PHC | 0.038 | 0.022 | 42 | 4.31 | 151 | 150 | 8 | R |
| | 0.001 | 0.001 | 91 | 4.04 | 101 | 146 | 9 | L |
| PHC > PRC | 0.008 | 0.005 | 61 | 4.15 | 150 | 131 | 10 | R |
| | 0.001 | 0.002 | 87 | 3.53 | 108 | 136 | 11 | L |
| Experiment 2 | | | | | | | | |
| PRC seed | <0.001 | <0.001 | 777 | 4.82 | 109 | 152 | 20 | L |
| | <0.001 | <0.001 | 849 | 4.73 | 149 | 145 | 10 | R |
| PHC seed | <0.001 | <0.001 | 669 | 5.31 | 147 | 137 | 13 | R |
| | <0.001 | <0.001 | 637 | 4.91 | 107 | 139 | 14 | L |
| PRC > PHC | <0.001 | <0.001 | 167 | 4.70 | 105 | 153 | 15 | L |
| PHC > PRC | 0.022 | 0.024 | 66 | 4.09 | 153 | 130 | 9 | R |
| | 0.047 | 0.026 | 53 | 3.74 | 108 | 136 | 18 | L |

Entorhinal subregions showing significant functional connectivity (one-sample t-test) or differential connectivity (paired t-test) with bilateral PRC or PHC seeds (Z > 2.3, $p_{cluster}$ <0.05, $N_{Exp. 1}$ = 15, $N_{Exp. 2}$ = 14). Single-subject beta maps were normalized on the group-specific T1-template and masked with a manually defined EC ROI. The EC covered 26 coronal slices on the template (y = 154: most anterior slice, y = 129: most posterior slice), with coronal slices being oriented orthogonal to the hippocampal long-axis. See also *Figure 1* and *Figure 1—figure supplement 1*.

running from anterior-ventral-lateral to posterior-dorsal-medial EC. In addition, we plotted functional connectivity preference of unilateral (left and right) PRC/PHC seeds with both the ipsi- and contralateral EC to evaluate whether connectivity patterns were symmetric across hemispheres (see *Figure 2—figure supplement 1B* for results of data set 1). These additional analyses confirmed the findings of bilateral connectivity analyses with an anterior-posterior and lateral-medial gradient of EC connectivity for contralateral PRC and PHC seeds. These findings also suggest that the dissociation of EC connectivity cannot be simply explained by local autocorrelations of neighboring voxels.

Our next analyses assessed the reliability of topographic differences in EC connectivity with PHC and PRC. If the topographic organization is reliable across participants, then it should be possible to predict the connectivity preference of specific EC voxels within any participant, simply by knowing the connectivity preferences of corresponding voxels in other participants. To test this strong prediction, we conducted a multivariate pattern classification analysis using a leave-one-subject-out cross-validation scheme. Specifically, we trained a multivariate support vector machine classifier (see 'Materials and methods' section) on data from all but one subject, entering only the x-, y-, and z-coordinates of each voxel and the relative preference of PRC and PHC connectivity (based on the subject's paired t-test t-maps, analogous to the definition of preference in the previous section). The classifier was then tested on the remaining subject, and the accuracy of this validation step was calculated as the proportion of EC voxels that were classified correctly as being preferentially connected to PRC or PHC in the tested subject. The training and cross-validation steps were repeated for all combinations of participants, and a mean classification accuracy score was computed. Classification accuracy was significantly above chance across both data sets (p < 0.001). Mean classifier accuracies across all EC voxels and subjects were around 60% (Exp. 1: left 62%, right 60%, Exp. 2: left 67%, right 57%). However, connectivity preference of voxels in the very anterior-lateral and posterior-medial EC could be predicted with more

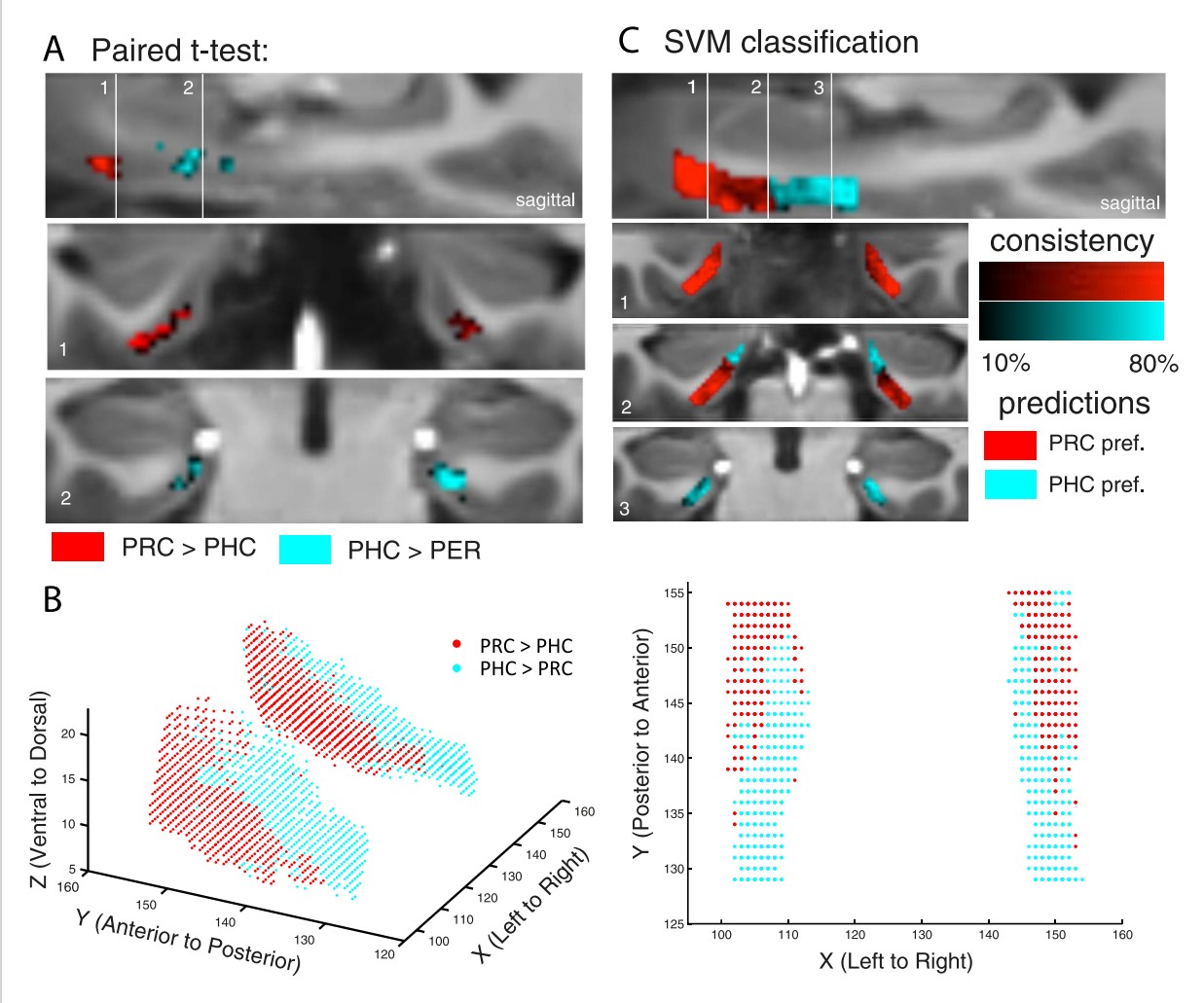

**Figure 2**. Differential connectivity topography of PRC vs PHC seeds with the EC for Experiment 1. (**A**) To assess differential connectivity of PRC vs PHC with the EC, voxelwise paired-sample t-tests were performed on the normalized single-subject beta maps (resulting from seed-to-voxel connectivity analyses). Significant clusters for Exp. 1 are shown for two coronal sample slices ($Z > 2.3$. $p_{cluster} < 0.05$, $N_{Exp. 1} = 15$) at the level of the anterior (1) and posterior (2) hippocampal head. (**B**) To visualize the 3-dimensional geometry of connectivity, the connectivity preference with PRC vs PHC of each EC voxel was plotted along the x-, y-, and z-axis (red: $T_{PRC > PHC} > 0$, blue: $T_{PHC > PRC} > 0$). Axes terminology is relative to the long-axis of the hippocampus. (**C**) Classification of PRC vs PHC connectivity preference was tested across subjects based on the x-y-z coordinate of an EC voxel. Multivariate classification (support vector machine; leave-one-subject-out cross-validation) was significant across both data sets ($p < 0.001$; accuracies: Exp. 1: left: 62%, right: 60%, Exp. 2: left: 67%, right: 57%), which confirms a spatial dissociation of entorhinal connectivity with PRC vs PHC. Predicted clusters are color-coded in red vs. blue, bright regions denote high consistency of the classifier (accuracy for each voxel across subjects). Results are shown for Exp. 1. See also *Figure 2—figure supplement 1* for 3D plots of Exp. 2 and for unilateral seeds of Exp. 1.

The following figure supplement is available for figure 2:

**Figure supplement 1**. Differential connectivity topography of bilateral (**A**) and unilateral (**B**) PRC vs PHC seeds with the EC for Experiment 2 (**A**) and Experiment 1 (**B**).

than 80% accuracy. These analyses confirm a spatial dissociation in connectivity between EC subregions with regard to PRC vs PHC seeds. Predicted clusters of preferential connectivity are color-coded in *Figure 2C* (bright regions denote high consistency of the classifier).

Moreover, we repeated the classification analysis on the combined data of both studies to predict al-EC and pm-EC functional subregions (based on preferential connectivity with PRC vs PHC) across all participants. Anatomical landmarks for these clusters are described further below in 'Landmarks for delineation of al-EC and pm-EC' and al-EC and pm-EC masks in template and MNI space are available

**Video 1.** 3D animation of entorhinal, subicular and parahippocampal subregions.

online (*Source codes 1, 2*). Moreover, predicted al-EC and pm-EC subregions as well as the subicular and parahippocampal subregions are shown in a 3D animation in *Video 1*.

## PRC and PHC show significantly different connectivity patterns along the transverse and longitudinal axis of the EC

Anatomical studies in rodents have demonstrated a rostrolateral-caudomedial dissociation of EC connectivity with PRC vs PHC, and data in nonhuman primates suggest a similar gradient of differential connectivity along anterior-posterior and lateral-medial axes (*Suzuki and Amaral, 1994*; *Witter et al., 2000a*; *van Strien et al., 2009*). In order to directly test for significant differences of PRC vs PHC intrinsic functional connectivity between the anterior vs posterior and lateral vs medial EC, we divided the template EC ROI equally into four portions: anterior-lateral, anterior-medial, posterior-lateral and posterior-medial (see *Figure 3A*, right panel). Mean parameter estimates for PRC and PHC connectivity (mean betas) were extracted across all voxels in each section for each subject. Note that the lateral-medial split was performed for each coronal slice individually as the EC is curved along the longitudinal axis and furthermore, that this separation also corresponds to a ventral-dorsal split.

Repeated-measures ANOVAs with PRC and PHC connectivity estimates revealed significant two-way interactions of seed region × anterior-posterior EC section (Exp. 1: $F(1,14) = 56.0$, $p < 0.001$; Exp. 2: $F(1,13) = 95.9$, $p < 0.001$) and seed region × lateral-medial EC section (Exp. 1: $F(1,14) = 11.3$, $p = 0.005$; Exp. 2: $F(1,13) = 32.8$, $p < 0.001$).

Follow-up paired sample t-tests confirmed significantly greater PRC than PHC connectivity with the anterior EC (Exp. 1: $t(14) = 5.3$; $p < 0.001$; Exp. 2: $t(13) = 4.5$, $p = 0.001$) and vice versa significant higher PHC than PRC connectivity with the posterior EC (Exp. 1: $t(14) = 4.7$; $p < 0.001$; Exp. 2: $t(13) = 2.6$; $p = 0.019$). With regard to the lateral-medial dissociation, the PRC showed significantly higher intrinsic functional connectivity than the PHC with the lateral EC section (Exp. 1: $t(14) = 5.2$; $p < 0.001$; Exp. 2: $t(13) = 3.8$, $p = 0.002$). However, there was no significant difference in connectivity between seeds with the medial section of the EC (all p-values > 0.80). Although connectivity estimates did not differ in the medial EC portion, the PHC showed significantly stronger connectivity with medial than lateral EC (Exp. 1: $t(14) = 4.4$; $p < 0.001$; Exp. 2 $t(13) = 2.6$, $p = 0.021$). There were no significant three-way interactions between seed, longitudinal and sagittal EC section (all p-values > 0.25).

To characterize the topography of PRC and PHC functional connectivity with voxels along the longitudinal (anterior to posterior) and transverse (lateral to medial) axis of the EC, we plotted mean parameter estimates for each slice (see *Figure 3B*). These slice-by-slice plots further confirmed a dominant topographical organization of EC connectivity along the longitudinal axis with decreasing PRC connectivity and increasing PHC connectivity from anterior to posterior to EC. Furthermore, lateral to medial connectivity plots demonstrated decreasing PRC and increasing PHC connectivity.

## Subicular connectivity profiles related to functional EC subregions and PRC/PHC

### EC subregions differentially interact with proximal and distal subiculum

Anatomical studies in rodents have demonstrated that LEC and MEC exhibit different patterns of connectivity along the proximo-distal (transverse) axis of the subiculum and CA1 (e.g., *Witter et al., 2000a*). Similarly, anterolateral vs posteromedial EC regions in nonhuman primates have been shown to exhibit differential connectivity with proximal vs distal subiculum and CA1 (*Witter and Amaral, 1991*). These findings motivated us to test whether functional connectivity between our functional EC subregions and the hippocampus is topographically organized. Within the hippocampus, we focused on the subiculum, based on previous findings demonstrating that the topographic organization of cortico-hippocampal functional connectivity is most prominent in the subiculum (*Libby et al., 2012*).

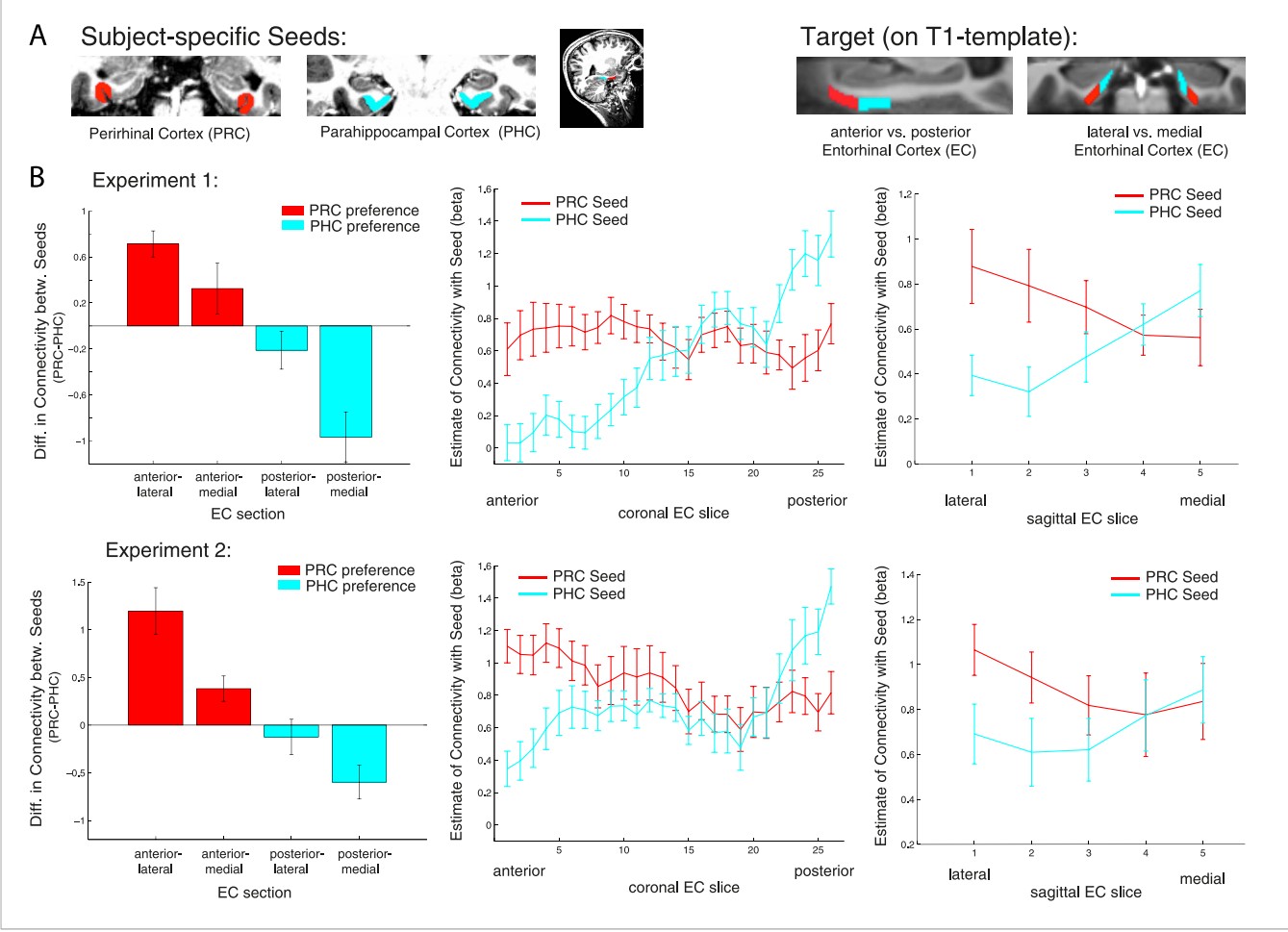

**Figure 3**. Anterior-posterior and lateral-medial gradients of entorhinal connectivity with PRC vs PHC seeds. (**A**) To test for an anterior-posterior or lateral-medial dissociation of EC connectivity with PRC vs PHC seeds (upper panel, left), we divided the EC template mask into four equal portions (upper panel, right) and extracted mean parameter estimates (betas) from each subsection. (**B**) Repeated-measures ANOVAs revealed significant seed (PRC vs PHC) × anterior-posterior EC section and seed × lateral-medial EC section interactions (p < 0.001 for both data sets; $N_{Exp1}$ = 15, $N_{Exp2}$ = 14). Slice-by-slice plots of connectivity estimates along the longitudinal and transverse EC axis confirmed an anterior-to-posterior and lateral-to-medial dissociation with decreasing PRC-connectivity and increasing PHC-connectivity. As the number of sagittal EC slices differed from anterior to posterior, we divided each coronal EC slice into 5 equal portions (with 1 being most lateral and 5 most medial EC) and calculated mean betas for each portion.

Consistent with our previous analyses for the EC, we divided the subiculum into four equal portions along the longitudinal (anterior vs posterior) and transverse (lateral ['proximal'] vs medial ['distal']) axis. We then extracted mean parameter estimates of functional connectivity with anterior-lateral and posterior-medial EC for each subicular section and each subject and submitted them to a factorial ANOVA to test for connectivity differences as a function of EC subregion (al-EC vs pm-EC) and anterior vs posterior and proximal vs distal subiculum subregions. Similar to the division of the EC, we performed the proximal-distal cut individually for each coronal subiculum slice.

Repeated-measures ANOVAs with EC seed connectivity estimates revealed a significant two-way interaction of seed EC subregion × proximal vs distal subiculum (Exp. 1: F(1,14) = 25.7, p < 0.001; Exp. 2: F(1,13) = 24.3, p < 0.001). However, there was no interaction between EC seed region and longitudinal subiculum sections (p > 0.26). Follow-up t-tests confirmed that the proximal subiculum showed significantly greater connectivity with al-EC than with pm-EC (Exp. 1: t(14) = 2.27; p = 0.040; Exp. 2: t(13) = 2.15; p = 0.049) and, conversely, that distal subiculum showed significantly greater connectivity with pm-EC than with al-EC (Exp. 1: t(14) = 3.06; p = 0.008; Exp. 2: t(13) = 4.24, p = 0.001). Slice-by-slice plots confirmed that connectivity of the al-EC decreased from proximal (lateral) to distal

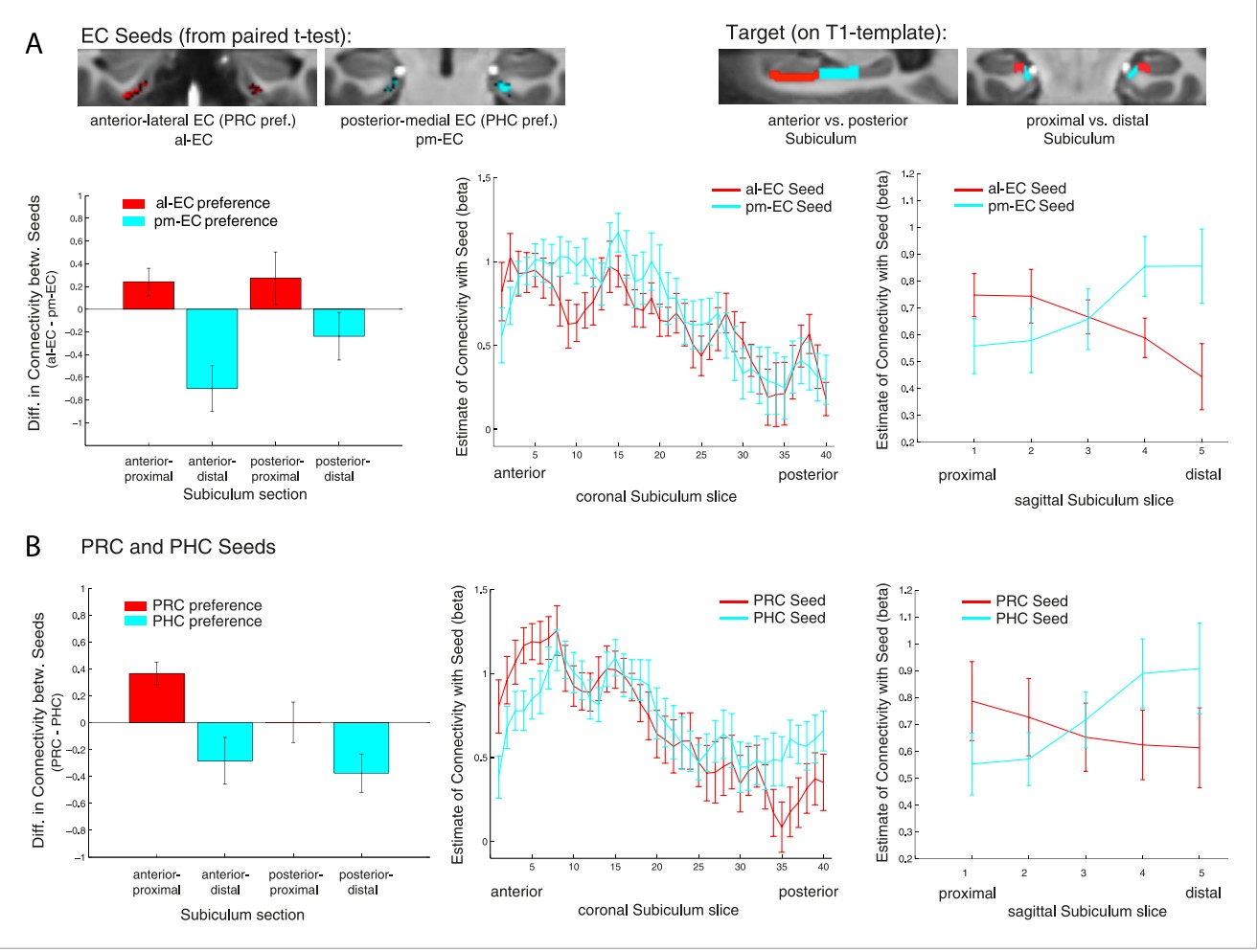

**Figure 4**. Functional connectivity gradients in the subiculum related to EC subregions and PRC/PHC seeds. (**A**) To test for differential connectivity of EC functional subdivisions with the subiculum, anterior-lateral EC (al-EC) and posterior-medial EC (pm-EC) regions that exhibited preferential connectivity with PRC vs PHC, respectively (see paired t-tests in *Figure 2A*) were used as seed regions. The subiculum ROI was equally divided into four portions along the longitudinal (anterior vs posterior) and transverse (proximal vs distal) axis and mean betas of functional connectivity with EC seeds extracted for each subsection. Repeated-measures ANOVAs revealed a significant seed (al-EC vs pm-EC) × proximal-distal subiculum interaction in both datasets ($p < 0.001$; $N_{Exp1} = 15$, $N_{Exp2} = 14$; results shown for Exp. 1). Slice-by-slice plots of connectivity estimates demonstrated decreasing al-EC-connectivity and increasing pm-EC connectivity from proximal to distal subiculum but no anterior-posterior dissociation. (**B**) Similarly, connectivity for PRC vs PHC seeds with the subiculum along the longitudinal and transverse axis was evaluated. Seed (PRC vs PHC) × proximal-distal subiculum section interactions were significant across both datasets ($p < 0.01$) with preferential connectivity of PRC with proximal and PHC with distal subiculum, respectively. Slice-by-slice plots of connectivity estimates along the hippocampal long axis revealed stronger PRC connectivity with the most anterior and stronger PHC connectivity with the most posterior subiculum (= 8 slices), respectively (Exp. 1). See also *Figure 4—figure supplement 1* for data of Exp. 2.

The following figure supplement is available for figure 4:

**Figure supplement 1**. Functional connectivity gradients in the subiculum related to EC subregions (**A**) and PRC/PHC seeds (**B**) for Experiment 2.

(medial) subiculum, whereas pm-EC connectivity increased (see *Figure 4A* and *Figure 4—figure supplement 1A* for data of Experiment 1 and 2, respectively).

## Repeated-measures ANOVAs and slice-by-slice plots for PRC/PHC seeds

Previous human resting-state fMRI studies at 3 Tesla (*Kahn et al., 2008*; *Libby et al., 2012*) have reported reliable differences in connectivity between the PRC and PHC with the hippocampus along the longitudinal hippocampal axis, most prominently with the subiculum (*Libby et al., 2012*). However, a dissociation of PRC vs PHC connectivity along the proximo-distal axis of the subiculum,

as demonstrated between PRC and POR in rodents (*Naber et al., 1999*, *2001*; *Agster and Burwell, 2013*), has not been reported so far in humans.

In order to examine whether PRC vs PHC seeds show differential intrinsic functional connectivity with the subiculum along the transverse and/or longitudinal axis, we also tested for significant interactions between PRC vs PHC connectivity with anterior vs posterior or proximal vs distal subiculum. Repeated-measures ANOVAs with PRC and PHC connectivity estimates revealed a significant two-way interaction of seed region × proximal vs distal subiculum (Exp. 1: $F(1,14) = 20.2$, $p = 0.001$; Exp. 2: $F(1,13) = 14.4$, $p = 0.002$). However, the interaction between seed and anterior vs posterior subiculum was not significant (Exp. 1: $F(1,14) = 2.5$, $p = 0.135$; Exp. 2: $F(1,13) = 1.4$, $p = 0.257$).

Follow-up t-tests confirmed significantly greater PRC than PHC connectivity with the proximal subiculum (Exp. 1: $t(14) = 3.7$; $p = 0.002$; Exp. 2: $t(13) = 3.9$, $p = 0.002$). With regard to distal subiculum there was significant higher PHC than PRC connectivity in Experiment 1 ($t(14) = 3.1$, $p = 0.009$) and a trend towards a difference in Experiment 2 ($t(13) = 2.0$; $p = 0.063$).

Slice-by-slice plots further confirmed a dissociation of PRC/PHC connectivity along the transverse hippocampal axis with decreasing PRC connectivity and increasing PHC connectivity from proximal to distal subiculum (see *Figure 4B* and *Figure 4—figure supplement 1B* for data of Experiment 1 and 2, respectively). Although the interaction between PRC vs PHC seed and anterior-posterior subiculum was not significant, slice-by-slice extractions suggested that differential connectivity may be present in the most anterior and most posterior slices of subiculum. Additional analyses indeed revealed that PRC connectivity was significantly greater than PHC connectivity within the most anterior 8 slices of subiculum (Exp. 1: $t(14) = 2.5$; $p = 0.022$; Exp. 2: $t(13) = 3.9$, $p = 0.002$), while PHC connectivity was significantly higher than PRC connectivity within the most posterior 8 slices in Experiment 1 ($t(14) = 2.3$; $p = 0.035$ ), but not in Experiment 2 ($t(13) < 1$, $p > 0.4$).

However, we note that we did not segment the subiculum within the hippocampal tail (where anatomical borders are difficult to delineate) and thus might underestimate any difference in the posterior subiculum.

## The topographic organization of PRC and PHC connectivity with subiculum differs from that of al-EC and pm-EC with subiculum

The above analyses revealed that the connectivity differences between al-EC and pm-EC along the transverse axis of the subiculum parallel the differences between PRC and PHC. However, only PRC and PHC seeds showed dissociable connectivity gradients along the longitudinal axis of the subiculum, whereas no such difference was seen between al-EC and pm-EC. This suggests that the gradients of al-EC and pm-EC connectivity with subiculum are not merely a reflection of their differential connectivity with PRC and PHC. To quantify the differences between the connectivity profiles of entorhinal functional subregions and PRC/PHC along the long axis of the subiculum, we calculated differences between al-EC/pm-EC and PRC/PHC connectivity estimates for anterior and posterior subiculum subsections. Repeated-measures ANOVAs revealed a significant interaction between seed regions (Δal-EC/pm-EC vs ΔPRC/PHC) × longitudinal subiculum section (anterior vs posterior) for Experiment 1 ($F(1,14) = 7.3$, $p = 0.017$) with a similar trend evident in the dataset of Experiment 2 ($F(1,13) = 3.4$, $p = 0.087$). These additional analyses show that functional connectivity profiles of PRC vs PHC and al-EC vs pm-EC with the subiculum differ along the longitudinal hippocampal axis.

### Landmarks for delineation of al-EC and pm-EC

We used the multivariate classification approach to predict al-EC and pm-EC subregions across all subjects. Based on the relative connectivity preference of an EC voxel across subjects the classifier predicts PRC or PHC connectivity preference for the left out subject. Thereby we can compute consistency maps that show the consistency of predictions for each EC voxel across all subjects. This revealed regions of high and low consistency. Based on the predictions of the classifier we created a PRC-connectivity preference ('al-EC') and a PHC-connectivity preference EC ('pm-EC') mask. To provide these masks in a more usable manner (not on a partial volume T1-template), we created a whole brain high-resolution T1-template based on the MPRAGEs of all participants ($N = 29$; voxel size: 0.6 mm$^3$ isotropic, AC-PC aligned). We then aligned the EC masks on the whole-brain template (linear registration) and manually corrected outer borders, if these did not fit perfectly.

Below, we further describe the approximate boundaries of al- and pm-EC subregions based on coronal slices, moving from anterior to posterior on the whole brain template. These landmarks can be used for manual delineation of EC functional subregions. Notably, coronal slices on the whole-brain template are not orthogonally aligned to the hippocampal long-axis and have a different slice thickness compared to the partial-volume template on which results/figures are described. Furthermore, borders between al-EC and pm-EC differed slightly between hemispheres and thus landmarks are an intermediate approximation between sides.

At the most anterior level of the EC (when the amygdala is visible) EC is fully covered by al-EC. Moving posteriorly, the hippocampal head (HH) starts. Around 3–4 slices (~2 mm) after the first appearance of the HH, pm-EC appears at the very medial/dorsal tip of the EC, touching the amygdala. The border between pm-EC and al-EC approximately corresponds to the uncal notch, such that pm-EC covers the gyrus ambiens. Moving more posteriorly the border between pm-EC and al-EC moves progressively down (more lateral/ventral) until al-EC and pm-EC are around equal size at the level of two thirds of HH. At this level the amygdala has fully disappeared. Going further posteriorly the border between pm-EC and al-EC moves progressively more lateral/ventral, such that pm-EC also covers parts of the ventral/lateral EC half. At the level where the uncus separates from the HC (and only the fimbria is attached to both), the EC is almost fully covered by pm-EC. We delineated the EC until the collateral sulcus (or rhinal fissure) disappeared, which was one slice (0.6 mm) after disappearance of the uncus. Approximately the four most posterior EC slices (~2.4 mm) were fully covered by pm-EC.

Al-EC and pm-EC masks as well as the whole brain T1-template are available online in original (*Source code 1*; resolution: 0.6 mm$^3$ isotropic) and MNI space (*Source code 2*; resolution: 2 mm$^3$ isotropic). Furthermore, location of al-EC and pm-EC in relation to PRC, PHC and proximal vs distal subiculum is visualized in *Video 1*.

## Discussion

We report the first detailed topographic parcellation of the human EC on the basis of its functional connectivity with neocortical and hippocampal subregions. In two independent samples, our analyses revealed that anterior-lateral and posterior-medial EC subregions (al-EC and pm-EC, respectively) exhibited distinct patterns of intrinsic functional connectivity with regions in the neocortex (PRC and PHC) and hippocampal formation (subiculum). Specifically, the al-EC region could be delineated on the basis of preferential connectivity with PRC, whereas the borders of pm-EC were derived from connectivity with PHC. Al-EC and pm-EC, in turn, were found to have preferential connectivity with proximal and distal subiculum, respectively. Moreover, the pattern of subiculum connectivity with al-EC and pm-EC was partially distinct from its connectivity with PRC and PHC. A schematic summary of functional connectivity gradients in the subiculum related to PRC/PHC seeds and EC subdivisions is illustrated in *Figure 5*. These results reveal the functional topography of the human EC as a gateway between neocortex and hippocampus and show remarkable accordance with principles known from anatomical studies of rodents (rostrolateral vs caudomedial; for reviews see *Witter et al., 2000a*; *van Strien et al., 2009*) and studies of nonhuman primates (anterolateral vs posteromedial; see e.g., *Witter and Amaral, 1991*; *Suzuki and Amaral, 1994*). As we describe below, these data provide a link between basic and translational research on the human medial temporal lobes (*Small et al., 2011*; *Ranganath and Ritchey, 2012*) and results from detailed circuit level analyses of the rodent hippocampal formation (e.g., *Moser and Moser, 2013*).

Previous fMRI studies have used functional connectivity analyses on data collected at 3T to characterize topographic patterns of connectivity between the PRC, PHC, and hippocampal subfields (*Lacy and Stark, 2012*; *Libby et al., 2012*). These studies have generally found that PRC and PHC exhibit different patterns of connectivity along the longitudinal axis of the hippocampus. Unfortunately, these studies could not address the topographic organization of connectivity within the EC, possibly due to limitations in resolution and SNR. The present results demonstrate that the enhanced resolution and sensitivity of ultra-high field fMRI can overcome these limitations and reveal fine-grained topographical patterns in connectivity. Three-dimensional plots of entorhinal connectivity preferences revealed a gradient of decreasing PRC and increasing PHC connectivity running from anterior-lateral to posterior-medial EC. It is notable that, by training a pattern classifier on the coordinates of EC voxels that showed preferential connectivity with PRC or PHC within a subset of participants, we could reliably predict these voxels in the remaining participant. This finding indicates that the topography of neocortical connectivity within the EC is highly

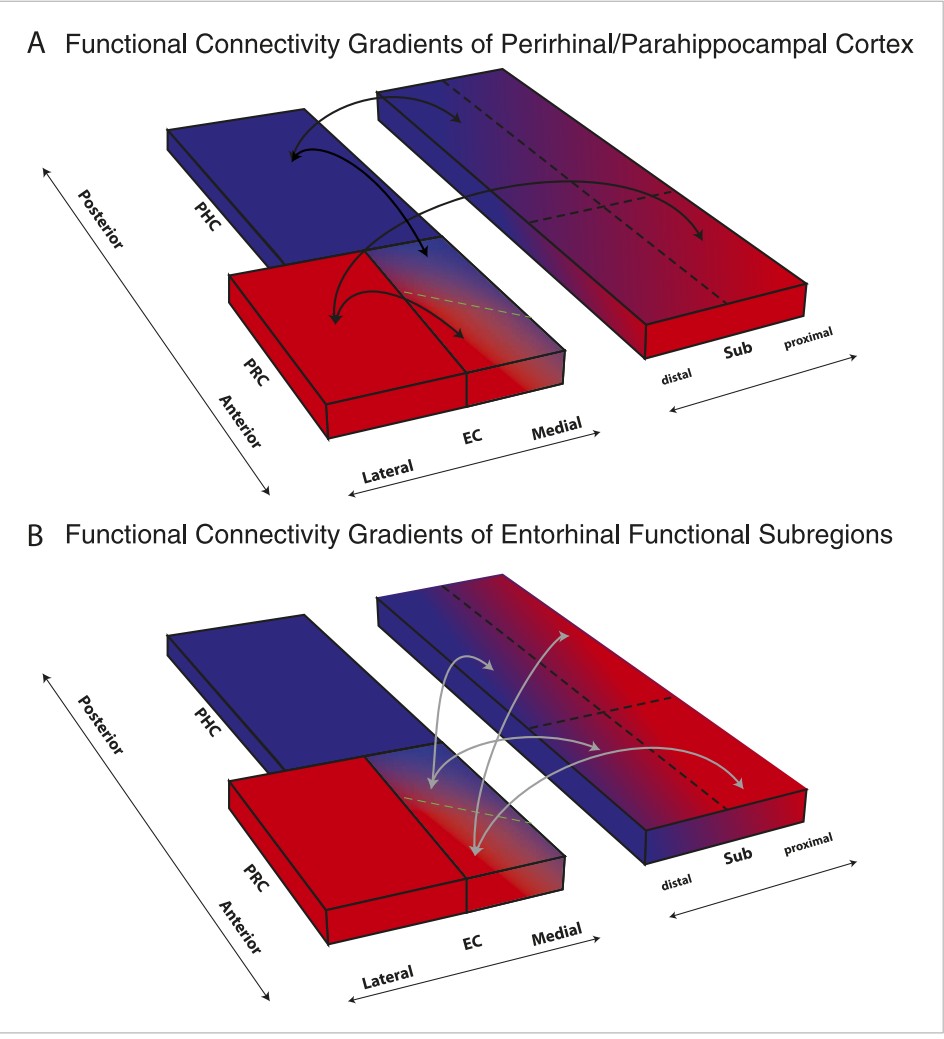

A  Functional Connectivity Gradients of Perirhinal/Parahippocampal Cortex

B  Functional Connectivity Gradients of Entorhinal Functional Subregions

**Figure 5**. Schematic summary of functional connectivity gradients in the subiculum related to PRC/PHC seeds and EC subdivisions. (**A**) Functional connectivity analyses revealed preferential connectivity of PRC (red) with the anterior-lateral EC and PHC (blue) with the posterior-medial EC. Regarding the subiculum, PRC showed strongest connectivity with most anterior and proximal parts, whereas PHC showed strongest connectivity with most posterior and distal parts of the subiculum. (**B**) Anterior-lateral (red) and posterior-medial (blue) EC exhibited a similar dissociation in connectivity with the subiculum along its transverse (proximal-distal) axis but there was no trend for a dissociation of entorhinal connectivity along the longitudinal axis of the subiculum.

conserved across participants, which, in turn, could indicate fundamental functional differences between the two EC subdivisions.

Two recent fMRI studies reported evidence for task-related activation differences between lateral and medial sections of EC in humans (*Schultz et al., 2012*; *Reagh and Yassa, 2014*). Schultz and colleagues (2012) reported differential activation in medial and lateral sections of EC during scene and face processing in a working memory task. Reagh and Yassa reported preferential activation in a medial section of EC during mnemonic discrimination of spatial locations and preferential activation in a lateral section of EC during mnemonic discrimination of objects. This functional dissociation was observed by splitting the EC into equally-sized lateral and medial parts according to the rodent terminology of 'LEC' and 'MEC'. Notably, they also found a trend towards a dissociation between anterior and posterior EC after a similar equal division along the longitudinal axis. Our data help to explain these findings by empirically demonstrating that al-EC and pm-EC exhibit differential functional connectivity with PRC and PHC. Numerous fMRI studies have shown that PHC is preferentially engaged in memory tasks that involve scenes, spatial or context information, whereas

PRC is preferentially engaged in memory tasks that involve object or item information (see *Ranganath and Ritchey, 2012*; *Ritchey et al., in press*; for review). Thus, it makes sense that EC subregions that interact predominantly with PRC or PHC also differentially participate in item and context processing. However, our data also suggest that a simple lateral-medial distinction does not capture the functional organization of EC. Future fMRI studies (as well as structural MRI studies, e.g., *Khan et al., 2014*) could more effectively study the EC by using the high-consistency pm-EC and al-EC masks derived from our data (see also 'Landmarks for delineation of al-EC and pm-EC'), or by using functional connectivity metrics to identify subject-specific EC subregions.

Concurrent with our investigation, Navarro Schröder and colleagues (*Navarro Schröder et al., 2015*) also studied the functional organization of the human EC. Whereas the present study focused on MTL connectivity, their study focused on functional connectivity of the EC with large-scale cortical networks. Consistent with our study, they also differentiated between anterior-lateral and posterior-medial EC subregions based on differential global network connectivity in resting state and task fMRI data (*Navarro Schröder et al., 2015*). They found that the al-EC exhibited stronger connectivity with regions in an anterior-temporal cortical system including medial-prefrontal and orbitofrontal cortex, whereas the pm-EC exhibited stronger connectivity with regions in a posterior-medial system including regions in occipital and posterior-parietal cortex. Analyses of task fMRI data revealed that al-EC activity was enhanced during processing of object information and pm-EC activity was enhanced during processing of scenes. Their findings are consistent with the idea that al- and pm-EC may be related to two cortico-hippocampal networks that support distinct types of memory (*Ranganath and Ritchey, 2012*; *Ritchey et al., in press*).

Although the EC is a major gateway for the hippocampus, neocortical regions such as PRC and PHC also have direct reciprocal connectivity with CA1 and subiculum (*Naber et al., 1999*, *2001*; *Agster and Burwell, 2013*). Our analyses revealed that the topographic differences in subicular connectivity with PRC vs PHC along the hippocampal transverse axis paralleled the differences of subicular connectivity with al-EC vs pm-EC. Whereas al-EC and PRC showed stronger connectivity with proximal subiculum, pm-EC and PHC showed stronger connectivity with distal subiculum. In contrast to the transverse axis, PRC/PHC vs al-EC/pm-EC connectivity profiles differed along the longitudinal hippocampal axis. For al-EC and pm-EC, there was no evidence or trend for an anterior-posterior dissociation, compatible with connectivity of LEC and MEC in rodents (*Naber et al., 1999*, *2001*; *Witter, 2006*; *O'Reilly et al., 2013*). In contrast, the most anterior subiculum showed stronger connectivity with PRC than PHC, whereas in one data set the most posterior subiculum (in the hippocampal body [HB]) showed stronger connectivity with PHC than PRC. This finding replicated the direct anatomical connectivity profiles observed in rodents (*Naber et al., 1999*, *2001*; *Agster and Burwell, 2013*). Such an anterior-posterior dissociation of hippocampal connectivity accords with findings from human resting-state fMRI studies that investigated functional connectivity profiles of PRC and PHC (rather than EC) with hippocampal subfields (*Libby et al., 2012*).

These functional connectivity data suggest that there might be two parallel cortico-hippocampal pathways in humans — one via the EC and one that is direct. The differences in the topographic organization of EC-subicular connectivity and PRC/PHC-subicular connectivity could have important functional implications. One implication is that the EC is not a simple anatomical extension of the PRC and PHC. If that were the case, we would not have observed any reliable difference between neocortical-hippocampal connectivity profiles and EC-hippocampal connectivity profiles. These results add support to the notion that the EC is more than a mere cortico-hippocampal relay (e.g., *Lavenex and Amaral, 2000*; *de Curtis and Paré, 2004*). One possibility is that this organization might allow a comparison between EC-gated hippocampal memory signals with direct neocortical input (e.g., *Naber et al., 1999*). Furthermore, the diffuse nature of LEC/MEC projections along the anterior-posterior hippocampal axis and a structured gradient of direct PRC/POR projections that has been identified in rodents could allow for integration of information across both processing streams (*Burwell, 2000*; *Witter et al., 2000b*; *Agster and Burwell, 2013*).

Results from the present study may be pertinent to understanding memory impairment in clinical conditions that compromise the structural integrity of the medial temporal lobes, including neurodegenerative diseases such as AD and frontotemporal lobar degeneration, temporal lobe epilepsy, depression, schizophrenia, developmental amnesia and ischemia. In AD, for instance, tau pathology emerges in lateral regions of the EC early in the course of the disease (*Braak and Braak, 1991*; *Braak and Del Tredici, 2004*). Analyses of functional connectivity can potentially reveal how EC

degeneration in the early stages of AD could impact the functional organization of distributed cortical networks (*Khan et al., 2014*; also see; *La Joie et al., 2014*) and also shed light on the transsynaptic progression of pathology in AD.

To summarize, the results of the present study provide a detailed description of the organization of functional connectivity within the human EC. Based on differential functional connectivity with PRC, PHC and subicular subregions, our data, along with those of Navarro Schröder et al. (in review), demonstrate that the human EC can be reliably subdivided into anterior-lateral and posterior-medial subregions that could be critical nodes in two cortico-hippocampal processing pathways. Future studies can apply the high resolution functional connectivity analyses to differentiate the roles of al-EC and pm-EC in memory and alterations of EC connectivity in AD and other neurodegenerative diseases.

## Materials and methods

### Participants

Two independent samples of 21 and 22 young, healthy subjects underwent high resolution fMRI scanning (Exp. 1: mean age 26 ± 3.6 yrs, 12 male; Exp. 2: mean age 28 ± 3.9 yrs, 7 male). Exclusion criteria were metallic implants (other than standard dental implants), tinnitus, known metabolic disorders or a history of neurological or psychiatric disorders. Both studies were approved by the local ethics committee of the University Magdeburg. All subjects gave written informed consent and consent to publish prior to participation and received monetary compensation for participation. Six subjects from Experiment 1 and six from Experiment 2 were excluded due to strong dropouts in the PRC and/or EC or due to severe movement artifacts. Functional connectivity analyses were performed on the residuals of task data after extraction of task effects ($N_{Exp.\ 1} = 15$, $N_{Exp.\ 2} = 14$).

### Tasks

#### Experiment 1 (encoding of novel vs familiar images of scenes)

During the fMRI session, subjects performed an incidental visual encoding paradigm. In 1 run, 120 new images (60 indoor and 60 outdoor), 60 'noise' images and 60 repetitions of one familiar image were presented randomly. The familiar image and the 'noise' images were familiarized using 10 repetitions each directly before the functional MR scan. Subjects made an indoor/outdoor judgment for each image by button press (for more details, see *Maass et al., 2014*).

#### Experiment 2 (Differentiation of original images and similar lures)

During fMRI data acquisition subjects performed an explicit mnemonic discrimination task. Stimuli were presented in short sequences consisting of 3–5 stimulus presentations of the same two stimuli (two similar versions of the same indoor scene). Sequences were presented in an event-related design. Each stimulus was presented for 3 s and stimuli were separated by a presentation of a scrambled noise picture for 3.5 s to prevent after-image or pop-out effects. Sequences were separated by a presentation of a fixation star for 4 s. Subjects had to keep the first stimulus of each sequence (target) in mind and indicate the third presentation of this exact stimulus via button press with their right index finger.

### Data acquisition and preprocessing

MRI data were acquired using a 7T MR system (Siemens, Erlangen, Germany). A 32-channel head coil was used (Nova Medical, Willmington, MA). First, a high-resolution whole head MPRAGE volume (TE = 2.8 ms, TR = 2500 ms, TI = 1050 ms, flip angle = 5°, resolution 0.6 mm isotropic) was acquired.

Subsequently, the fMRI session was run. T2*-weighted gradient echo planar images (EPIs) were acquired with an isotropic resolution of 0.8 mm (28 axial slices, TE = 22 ms, TR = 2000 ms, FOV 205 mm, matrix 256 × 256, partial Fourier 5/8, parallel imaging with grappa factor 4, bandwidth 1028 Hz/Px, echo spacing 1.1 ms, echo train length 40, flip angle 90°). The slices were acquired in an odd–even interleaved fashion oriented parallel to the hippocampus long axis. The fMRI session comprised 1 run (13 min) with 370 EPI volumes in Experiment 1 and 4 runs (13.5 min each) with 400 EPI volumes in Experiment 2. EPIs were distortion corrected using a point spread function mapping method (*In and Speck, 2012*) and motion corrected during the online reconstruction.

Finally, a high-resolution partial structural volume was acquired (T2*-weighted imaging, TE = 18.5 ms, TR = 680 ms, resolution 0.33 mm × 0.33 mm, 45 slices, slice thickness 1.5 mm + 25% gap, FOV 212 mm × 179 mm, matrix 640 × 540), with a slice alignment orthogonal to the hippocampus main axis.

Total MRI duration was around 60 min in Experiment 1 and 100 min in Experiment 2.

FMRI data pre-processing and statistical modeling was done in SPM8 (Wellcome Department of Cognitive Neuroscience, University College, London, UK). The pre-processing included slice timing correction and smoothing with a 1.5 mm full-width half-maximum Gaussian kernel (FWHM < 2 × voxel size) to keep high anatomical specificity. Outliers in average intensity and/or scan-to-scan motion were identified using the *ARTRepair toolbox* for SPM (Percent threshold in global intensity: 1.3, movement threshold: 0.3 mm/TR) and included as spike regressors.

To remove task effects, general linear models were run (including all task conditions and the movement parameters) and the residual images were saved for subsequent intrinsic functional connectivity analyses. Based on previous studies suggesting a linear superposition of task activity and spontaneous BOLD fluctuations, removing task-induced variance of event-related fMRI data should yield a remaining residual signal similar to 'continuous' resting state data (e.g., *Fox et al., 2006*). Although quantitative differences between residuals derived from task data and continuous resting state data have been reported (*Fair et al., 2007*), in qualitative terms, patterns of functionally connected regions have been shown to be remarkably consistent (*Fair et al., 2007*; *Lacy and Stark, 2012*).

## Segmentation of regions of interest

In order to analyze PRC vs PHC seed-to-voxel connectivity, we manually segmented PRC and PHC regions of interest (ROIs) for each subject on the individual high resolution MPRAGEs (which had been bias-corrected and coregistered to the individual mean EPIs). Furthermore, the EC and the subiculum were labelled on the T1-group template in order to analyze PRC vs PHC connectivity topography within the EC and al-EC vs pm-EC as well as PRC vs PHC connectivity topography within the subiculum, respectively, at group level (individual beta-maps were registered to the template). ROIs were traced on consecutive coronal slices bilaterally using MRIcron (Chris Rorden, Version 4 April 2011).

Tracing of the EC started anteriorly at the first slice where the amygdala was visible. Caudally the EC moves along the parahippocampal gyrus until the collateral sulcus disappeared (*Fischl et al., 2009*), typically at the level where also the HH ends, merging into the PHC. At anterior levels, the EC borders the amygdala nuclei medially (*Fischl et al., 2009*). When the gyrus ambiens disappears and the hippocampal fissure opens, the EC borders the parasubiculum medially. This EC-subiculum boundary was located at the angle formed by the most medial extent of both subiculum and EC (*Wisse et al., 2012*). The opening of the collateral sulcus typically coincides with the lateral border of the EC (*Fischl et al., 2009*), and was chosen as lateral limit. The PRC, which laterally abuts the EC, was defined as the region between the medial and lateral edges of the collateral sulcus (covering medial and lateral banks). Contrary to other markings schemes for the EC and PRC (*Insausti et al., 1998*), we did not mark the part of the EC within medial banks of the collateral sulcus that depends on the depth of the collateral sulcus, since this border shows remarkable within and between subject variability and is also sometimes difficult to identify due to partially occurring susceptibility artefacts. For the purpose of the current study, we further deleted PRC mask voxels that were directly neighbored to the EC/PRC border (leaving a gap of approx. 2 voxels ~1 mm) to avoid autocorrelations between PRC seed voxels and our target EC ROI. Segmentation of the PHC started one slice after the disappearance of the collateral sulcus, directly posterior to PRC and EC (approx. at the level where the HB starts). Labeling was continued posteriorly, ending on the last slice where the inferior and superior colliculi were jointly visible. The PHC was delineated as the region between subiculum (medial border) and the deepest point of the collateral sulcus (*Zeineh et al., 2001*).

The subiculum was labeled on the T1-template (0.8 mm isotropic resolution) in the HH and body (until the colliculi disappeared) according to the segmentation protocol of (*Wisse et al., 2012*). Note that we did not segment the hippocampal tail as borders were difficult to identify. For division of EC and subiculum masks into anterior-posterior and lateral-medial or proximal-distal subregions, see 'Second Level Analyses' (Univariate).

Subsequently, PRC and PHC masks were coregistered and resliced to the individual mean EPI images and manually adjusted to achieve a precise overlay on the functional data. To exclude voxels susceptible

to signal dropout, for each ROI, voxels with mean intensity (across timepoints) < 2 SD from the mean intensity (across voxels) in an ROI were removed from the ROI (*Libby et al., 2012*). Thresholding led to the rejection of no more than 5% of voxels in any ROI. Additionally, areas of PRC were occasionally subject to distortion artifact, and these voxels were manually deleted from ROIs. These adjusted and thresholded ROIs were used as seed regions for the functional connectivity analyses.

Probabilistic white matter (WM) and cerebral spinal fluid (CSF) masks were generated by automated segmentation (SPM8, 'New Segment') of the co-registered MPRAGE images and thresholded at p(tissue) > 0.95.

## First-level functional connectivity analyses

We performed seed-to-voxel correlational analyses on the native (preprocessed, unnormalized) residual fMRI data using the *conn-toolbox* (*Whitfield-Gabrieli and Nieto-Castanon, 2012*). First, functional connectivity patterns of PRC vs PHC seeds with the EC were analyzed. For each functional connectivity analysis, seed regions' average time series were generated as regressors of interest. As covariates of no interest, WM and CSF time series and subjects' realignment parameters (including spike regressors) were included to account for physiological noise and movements, respectively. Functional data were band-pass filtered for frequencies of 0.01–0.1 Hz. Bivariate correlations were computed, resulting in beta maps containing Fisher-transformed correlation coefficients. To perform group analyses, beta maps were registered to the group-specific T1 template (see below) and Z-standardized.

## Cross-participant alignment for group analyses

In order to enable precise cross-participant alignment for hippocampal and parahippocampal regions, we used Region of Interest-Advanced Normalization Tools (ROI-ANTS [*Klein et al., 2009*; *Yassa and Stark, 2009*; *Avants et al., 2011*]).

First, the Oxford Centre for Functional MRI of the Brain (FMRIB) software library (FSL 5.0.6 [*Smith et al., 2004*]) was used to register each single participant's structural MPRAGE and the mean functional EPI using epi_reg, a command-line program that belongs to the FMRIB's linear registration tool (FLIRT v6.0 [*Jenkinson and Smith, 2001*]) and was specifically written to register EPI images to structural images.

Second, a study-specific template was created (from individual MPRAGE images of Exp. 1) using the buildtemplateparallel.sh command-line script within ROI-ANTS (Cross-Correlation similarity metric [*Avants et al., 2010*]). Although the resulting alignment parameters already allow for a good registration to the template, we further improved normalization for the MTL regions by adding landmarks to the template. Therefore, the HH (on the first slice on which it appears), EC (on the first four consecutive slices, starting on the HH slice), the HB and the PHC (same slices as HB) were labeled on the T1-template as landmarks for the subsequent label-guided alignment. Similarly, subject-specific ROIs were drawn on the individual MPRAGEs to match the template priors.

Third, the expectation-based point set registration ('pse'; step size: SyN[0.5]) was used to register the individual MPRAGEs on the T1-template based on the labeled points sets (= MTL masks). The resulting transformation matrix was then applied to each participant's beta map as well as to the MTL masks in order to verify alignment precision. Finally, the aligned beta images were submitted to second-level group analyses.

## Second-level analyses

### Univariate

First, we calculated single-seed group connectivity maps using voxelwise one-sample t-tests to characterize the intrinsic connectivity profiles of PHC and PRC with EC (*Figure 1* and *Figure 1—figure supplement 1A*). Additionally, paired t-tests were performed to determine significant differences in PRC vs PHC connectivity (*Figure 2A*). Resulting t-maps were masked with the EC ROI and significant clusters determined by cluster-extent based thresholding ($Z > 2.3$; $p_{cluster} < 0.05$). In addition, we visualized the differential topographic pattern of PRC vs PHC connectivity along the x-y-z direction in three-dimensional plot of connectivity preference for each voxel (see *Figure 2B*). These analyses indicated relatively stronger connectivity of the PRC with the anterior-ventral-lateral EC and stronger connectivity of the PHC with the posterior-dorsal-medial EC.

In a next step, to directly test for significant differences between anterior/posterior and lateral/medial regions of the EC, we divided the template EC ROI into four equally-sized sections along the

longitudinal and horizontal (or transverse) plane, corresponding to coronal and sagittal cuts, respectively. As mentioned previously, the planes are defined in reference to the longitudinal axis of the hippocampus (which also corresponds to the EC long-axis). The lateral vs medial division was performed for each coronal slice individually in order to ensure equally-sized portions along the longitudinal axis. Mean parameter estimates were extracted from each of the four subsections for each subject (from the group-registered beta maps). The resulting PRC and PHC connectivity estimates were submitted to a 2 × 2 × 2 repeated-measures ANOVA with seed ROI (PHC vs PRC), longitudinal EC section (anterior vs posterior), and horizontal EC section (lateral vs medial) as factors. Furthermore, we assessed connectivity gradients across the longitudinal and horizontal axes by plotting slice-by-slice mean parameter estimates. As the number of transverse EC slices differed along the longitudinal axis (between 5 and 11 slices), we divided each coronal EC slice into 5 equal sections and calculated mean betas for each section. The same approach was used to test for differential connectivity of EC subregions and PRC/PHC with the subiculum along the longitudinal or transverse axis.

### Multivariate

Finally, we used a multivariate classifier (linear CSVMC) to further investigate the reliability of the topography of PRC vs PHC preferential connectivity within the EC. A linear support vector machine classifier (Linear CSVMC, *Chang and Lin, 2011*) was trained and tested on the x-y-z coordinates of all EC voxels across all subjects using PyMVPA 2.2.0 (*Hanke et al., 2009*). Independent data chunks were defined according to the individual subjects of each study. Each chunk consisted of the same amount of samples (EC voxels). For the purpose of evaluation of classification validity, a leave-one-subject-out cross-validation was performed. In each of the validation steps, a linear support vector machine was trained on the data of all but one subject and tested on the remaining one. Accuracy of the validation step was calculated as the proportion of the samples (voxels) that were classified correctly (as being preferentially connected to PRC or PHC). Overall classification accuracy was defined as the mean accuracy of all validation steps across subjects.

Mean classifier accuracy was tested for significance using non-parametric permutation testing. Over 1000 iterations, samples (EC voxels) of the training set were randomly relabeled and tested on the testing set using the same leave-one-subject-out cross-validation scheme as before, generating a non-parametric null distribution. Type I error rate was set at p < 0.001 based on this null distribution (*Nichols and Holmes, 2002*). Classification and permutation testing were performed separately for the left and right hemisphere and separately for Experiment 1 and Experiment 2. In order to investigate the consistency of correct classifications for each voxel, we calculated the proportion of correct classifications for each voxel across subjects. Voxels with high consistency are often correctly classified across subjects whereas voxels with low consistency are rarely classified correctly.

## Acknowledgements

We thank the Leibniz Institute for Neurobiology for providing access to their 7 Tesla MR scanner. Furthermore, we thank Selim Candan for assistance with the 3D animation and figures.

## Additional information

### Funding

| Funder | Grant reference | Author |
| --- | --- | --- |
| National Science Foundation (NSF) | Graduate Research Fellowship | Laura A Libby |
| John Simon Guggenheim Memorial Foundation | Fellowship | Charan Ranganath |
| University of Cambridge | Parke-Davis Exchange Fellowship | Charan Ranganath |
| Deutsche Forschungsgemeinschaft | SFB 776, TP A7 | Emrah Düzel |

The funders had no role in study design, data collection and interpretation, or the decision to submit the work for publication.

## Author contributions
AM, DB, Conception and design, Acquisition of data, Analysis and interpretation of data, Drafting or revising the article; LAL, CR, Analysis and interpretation of data, Drafting or revising the article; ED, Conception and design, Analysis and interpretation of data, Drafting or revising the article

## Author ORCIDs
David Berron, [iD] http://orcid.org/0000-0003-1558-1883
Laura A Libby, [iD] http://orcid.org/0000-0003-1484-4573
Charan Ranganath, [iD] http://orcid.org/0000-0001-5835-6091

## Ethics
Human subjects: The study was approved by the ethics committee of the University of Magdeburg. All subjects gave written informed consent, and consent to publish prior to participation and received monetary compensation for participation.

## Additional files

### Supplementary files
• Source code 1. Al-EC and pm-EC masks in template space and the corresponding high-resolution T1-weighted group template (0.6 mm isotropic resolution, whole brain, AC-PC aligned). Masks are predicted clusters derived by multivariate classification of PRC and PHC connectivity preference across both data sets.

• Source code 2. Al-EC and pm-EC mask in MNI space and the normalized T1-weighted group template (2 mm isotropic resolution).

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
