## [Decision Letter]

Thank you for sending your work entitled “Functional subdivisions of the human entorhinal cortex” for consideration at *eLife*. Your article has been favorably evaluated by Eve Marder (Senior editor) and three reviewers, one of whom is a member of our Board of Reviewing Editors.

All the individuals responsible for the peer review of your submission have agreed to reveal their identity: Howard Eichenbaum (Reviewing editor and peer reviewer), Menno Witter and Elizabeth Buffalo (peer reviewers).

The Reviewing editor and the other reviewers discussed their comments before we reached this decision, and the Reviewing editor has assembled the following comments to help you prepare a revised submission.

In this interesting and well-written study, the authors used 7 Tesla ultra-high field functional magnetic resonance imaging to identify functional subdivisions of the human EC through the analysis of preferred connectivity with parahippocampal and perirhinal cortices. Aside from a solidly supported hypothesis on how connectional patterns might allow the definition of functionally different domains in the human entorhinal cortex, the authors aimed to test other connectivity patterns in humans as well, including the perirhinal and parahippocampal-subicular projections. This study, as well as its counterpart study by Schröder et al., should be welcomed and heralded as an important step forward.

However, the reviewers were consensual in concern that this study fell into a well-known trap of comparative anatomical studies in that chosen nomenclatures are taken to have implications that have never been intended, neither implicitly nor explicitly. This leads to rather confounding representations of /comparisons with known data on the connectional organization of the parahippocampal region in different species. In addition, the study is hampered by addressing multiple questions, all relevant and worthy of analysis, which all and all are done very nicely by the authors, but the diversity adds to the confusion instead of counter-balancing this.

To start with the latter point, the Abstract does not fully represent the main findings of the study; while in the Abstract the differential connections of an anterolateral part of EC and a posteromedial part are described with PRC and PHC respectively and with the subiculum along its transverse axis, the data reported on the differential distribution along the anteroposterior axis of the subiculum of projections from PRC and PHC are not mentioned. This incomplete coverage is similarly found in the title. One wonders whether the latter data might not be moved to a separate paper; alternatively the title (preferentially) but certainly the Abstract should be changed.

The major problem as said above is the comparative part of the paper. Although the authors obviously aimed to cover a lot of experimental data on the organization of connectivity of the areas under investigation in rodents and monkeys they provide an incorrect presentation of the data with a few exceptions. In the Abstract we read that “anatomical studies have not reported such a distinction in nonhuman primates” which is at least incomplete and more likely incorrect. In the paper the authors refer to studies in the monkey showing differential connections along the proximodistal axis (Witter and Amaral) as well as connectional differences between PRC/PHC and EC (Suzuki and Amaral). The last sentence in the Abstract—“the first evidence that the human EC can be divided into functional subdivisions whose functional connectivity closely parallels the known anatomical connectivity patterns of the rodent lateral and medial EC”—is very much welcome but is in contrast to sentences in the body of the paper. Therefore the Abstract has to be rewritten.

The problem likely originates from a common problem with the interpretation of names/nomenclature. In the founding papers on EC, it has been proposed to divide this into 2 regions, based on cytoarchitectonic and additional hodological criteria. Authors initially used different names, but the field settled on lateral and medial entorhinal cortex. This has proven to be a very unfortunate nomenclature since the position of the entorhinal cortex on the cortical surface of the brain in many species is very different, while its connectivity is strikingly preserved. In rodents, the defined MEC has a posteromedial position, while LEC has an anterolateral position, so not that different from the findings in the human (AL-EC vs PM-EC). Based on these definitions, an enormous body of connectivity data has emerged that added defining features to LEC and MEC in rodents. In the monkey, authors have described areas that are cytoarchitectonically comparable to LEC and MEC in rodents, though authors tend to disagree a bit where they should be positioned precisely. So it all depends on definition. In the primate literature, both monkey and human, many authors tend to take it for granted that LEC means the laterally positioned part/half of the EC and MEC means that it is positioned medially, i.e. EC is divided into a lateral and a medial strip. Most known anatomical data in the monkey clearly show that this is incorrect (as it is in almost all other species!). MEC is likely the more posteriomedial part of EC and LEC is the more anterolateral portion, what has been problematic is to agree in the precise location of the border between them.

The following sentences in the Introduction clearly illustrate the problem: “In actuality, anatomical studies in nonhuman primates have not reported connectivity differences between MEC and LEC. Instead, these studies suggest an anterior-posterior gradient of EC connectivity, with the PRC tending to be interconnected with the anterior third of the EC, whereas the PHC tends to be interconnected with approximately the posterior two-thirds of the EC (42).” The former is correct if one defines LEC and MEC as two areas that are physically located lateral and medial, but very incorrect if one looks at the data, which are partially correctly summarized in the second sentence. In the Suzuki and Amaral paper the preferred connections of PHC to EC are in the posterolateral domain and those form PRC are in the anteromedial domain, strikingly similar to what the current data show in the human.

This sentence in the subsection “PRC and PHC show significantly different connectivity patterns along the transverse and longitudinal axis of the EC” further illustrates the confusion: “Anatomical studies in rodents have demonstrated a lateral-medial dissociation of EC connectivity with PRC vs PHC, while data in nonhuman primates suggest an anteriorposterior dissociation”. One can argue that the connectivity patterns are actually quite similar but slightly rotated in space.

At the start of the subsection headed “EC subregions differentially interact with proximal and distal subiculum”: “In nonhuman primates, anterior vs posterior EC regions have been shown to exhibit different patterns of connectivity with proximal and distal subiculum and CA1 (48).” The actual data show an anterolateral vs posteromedial organization.

There are several other instances where the chosen formulation obscures a clear line of thinking. The authors clearly think along similar lines when they write in the first paragraph of the Discussion: “These results reveal the functional topography of the human EC as a gateway between neocortex and hippocampus and show remarkable accordance with principles known from anatomical studies of rodents (lateral vs medial; for reviews see (51; 44) and studies of non-human primates (anterior vs posterior; see e.g. [42]; [48]).” Unfortunately in referring to the data of, for example Reagh and Yassa, matters become complicated again (third paragraph of the Discussion), since there they copy the nomenclature LEC and MEC from the authors which unfortunately is based on using LEC for a lateral part of EC and MEC for a medial part of EC. In the next sentence the authors than claim that their data demonstrate “that lateral and medial EC exhibit differential functional connectivity with PRC and PHC.” Here they refrain from using either LEC/MEC or their own adopted AL-EC and PM-EC.

We strongly suggest that the authors start by defining areas as AL-EC and PM-EC and not to use any indications of LEC and MEC unless clearly defined for each instance.

Other major concerns:

1) The analyses are rigorous and the results are strong. In order to make these data maximally useful for other researchers, it would be nice to include more detail about the location of the border between anterior-medial EC and posterior-lateral EC. For example, some textual description regarding landmarks that can be used to identify the border would be useful. Although the data (particularly the data shown in Figure 2), certainly support the existence of a medial-lateral distinction within EC, the accompanying movie shows mostly an anterior-posterior distinction. This potential inconsistency should be clarified.

2) A similar AP distinction was identified in a single neuron recording study by Killian et al. (Nature 2012).

3) Given the observations that highlight differential gradients rather than distinctly separate EC areas, is it appropriate to title the paper as revealing “functional subdivisions” of EC?

---

## [Author Response]

*The reviewers were consensual in concern that this study fell into a well-known trap of comparative anatomical studies in that chosen nomenclatures are taken to have implications that have never been intended, neither implicitly nor explicitly. This leads to rather confounding representations of /comparisons with known data on the connectional organization of the parahippocampal region in different species. In addition, the study is hampered by addressing multiple questions, all relevant and worthy of analysis, which all and all are done very nicely by the authors, but the diversity adds to the confusion instead of counter-balancing this*.

*To start with the latter point, the Abstract does not fully represent the main findings of the study; while in the Abstract the differential connections of an anterolateral part of EC and a posteromedial part are described with PRC and PHC respectively and with the subiculum along its transverse axis, the data reported on the differential distribution along the anteroposterior axis of the subiculum of projections from PRC and PHC are not mentioned. This incomplete coverage is similarly found in the title. One wonders whether the latter data might not be moved to a separate paper; alternatively the title (preferentially) but certainly the Abstract should be changed*.

We agree that our Abstract did not cover our findings in their entirety. We have changed the Abstract and now also report that connectivity of PRC/PHC seeds with subiculum differed from al-EC/pm-EC-subicular connectivity. We revised the following paragraph in the Abstract:

“In two independent datasets, PRC showed preferential intrinsic functional connectivity with anterior-lateral EC and PHC with posterior-medial EC. These EC subregions, in turn, exhibited differential connectivity with proximal and distal subiculum. In contrast, connectivity of PRC and PHC with subiculum followed not only a proximal-distal but also an anterior-posterior gradient.”

In addition, we also changed the title to: “Functional subregions of the human entorhinal cortex”.

*The major problem as said above is the comparative part of the paper. Although the authors obviously aimed to cover a lot of experimental data on the organization of connectivity of the areas under investigation in rodents and monkeys they provide an incorrect presentation of the data with a few exceptions. In the Abstract we read that “anatomical studies have not reported such a distinction in nonhuman primates” which is at least incomplete and more likely incorrect. In the paper the authors refer to studies in the monkey showing differential connections along the proximodistal axis (Witter and Amaral) as well as connectional differences between PRC/PHC and EC (Suzuki and Amaral). The last sentence in the Abstract—“the first evidence that the human EC* can *be divided into functional subdivisions whose functional connectivity closely parallels the known anatomical connectivity patterns of the rodent lateral and medial EC”—is very much welcome but is in contrast to sentences in the body of the paper. Therefore the Abstract has to be rewritten*.

Thank you very much for highlighting this. We have rewritten the first sentences in the Abstract which now read:

“Studies in rodents and nonhuman primates suggest that EC can be divided into subregions that connect differentially with perirhinal (PRC) vs parahippocampal cortex (PHC) and with hippocampal subfields along the proximo-distal axis.”

*The problem likely originates from a common problem with the interpretation of names/nomenclature. In the founding papers on EC, it has been proposed to divide this into 2 regions, based on cytoarchitectonic and additional hodological criteria. Authors initially used different names, but the field settled on lateral and medial entorhinal cortex. This has proven to be a very unfortunate nomenclature since the position of the entorhinal cortex on the cortical surface of the brain in many species is very different, while its connectivity is strikingly preserved. In rodents, the defined MEC has a posteromedial position, while LEC has an anterolateral position, so not that different from the findings in the human (AL-EC versus PM-EC). Based on these definitions, an enormous body of connectivity data has emerged that added defining features to LEC and MEC in rodents. In the monkey, authors have described areas that are cytoarchitectonically comparable to LEC and MEC in rodents, though authors tend to disagree a bit where they should be positioned precisely. So it all depends on definition. In the primate literature, both monkey and human, many authors tend to take it for granted that LEC means the laterally positioned part/half of the EC and MEC means that it is positioned medially, i.e. EC is divided into a lateral and a medial strip. Most known anatomical data in the monkey clearly show that this is incorrect (as it is in almost all other species!). MEC is likely the more posteriomedial part of EC and LEC is the more anterolateral portion, what has been problematic is to agree in the precise location of the border between them*.

We realize that we have misinterpreted the animal literature and we are very grateful for this clarification. As described in detail below we have revised our Introduction and Discussion accordingly. As we also think that this misunderstanding is a common problem in the human imaging community, we added a novel paragraph about the nomenclature of EC subdivisions in rodents to the Introduction:

“Notably, although terminology for EC subdivisions in the rat emphasize the lateral to medial axis, these areas do not differ solely with respect to their position in relation to the hippocampal formation and the rhinal fissure (51). In actuality, LEC occupies the rostrolateral portion of the EC, whereas MEC occupies the caudomedial portion of the EC.”

*The following sentences in the Introduction clearly illustrate the problem: “In actuality, anatomical studies in nonhuman primates have not reported connectivity differences between MEC and LEC. Instead, these studies suggest an anterior-posterior gradient of EC connectivity, with the PRC tending to be interconnected with the anterior third of the EC, whereas the PHC tends to be interconnected with approximately the posterior two-thirds of the EC (*[42]*).” The former is correct if one defines LEC and MEC as two areas that are physically located lateral and medial, but very incorrect if one looks at the data, which are partially correctly summarized in the second sentence. In the Suzuki and Amaral paper the preferred connections of PHC to EC are in the posterolateral domain and those form PRC are in the anteromedial domain, strikingly similar to what the current data show in the human*.

Thank you for this notification. We assume that there was a spelling error in the reviewers’ comment and he/she meant: “PHC to EC are in the posteromedial domain and those form PRC are in the anterolateral domain”. We revised the description of EC connectivity topography in nonhuman primates accordingly. In this respect, we also added the citation of [24] for evidence of a functional anterior-posterior gradient in monkeys. The following paragraph has been revised in the Introduction:

“In primates, ventral HC and the adjacent EC are situated in a relatively more rostral position […] single-unit recording study of grid-cell-like neurons in nonhuman primates (24).”

*This sentence in the subsection “PRC and PHC show significantly different connectivity patterns along the transverse and longitudinal axis of the EC” further illustrates the confusion: “Anatomical studies in rodents have demonstrated a lateral-medial dissociation of EC connectivity with PRC vs. PHC, while data in nonhuman primates suggest an anteriorposterior dissociation”. One* can *argue that the connectivity patterns are actually quite similar but slightly rotated in space*.

We agree and have thus changed that sentence:

“Anatomical studies in rodents have demonstrated a rostrolateral-caudomedial dissociation of EC connectivity with PRC vs PHC, and data in nonhuman primates suggest a similar gradient of differential connectivity along anterior-posterior and lateral-medial axes (42; 51; 44).”

*At the start of the subsection headed “EC subregions differentially interact with proximal and distal subiculum”: “In nonhuman primates, anterior vs. posterior EC regions have been shown to exhibit different patterns of connectivity with proximal and distal subiculum and CA1 (*[48]*).” The actual data show an anterolateral versus posteromedial organization*.

We apologize this misunderstanding, which we have corrected in the revised manuscript:

“Anatomical studies in rodents have demonstrated that LEC and MEC exhibit different patterns of connectivity along the proximo-distal (transverse) axis of the subiculum and CA1 (e.g. Witter et al., 2000). Similarly, anterolateral vs posteromedial EC regions in nonhuman primates have been shown to exhibit differential connectivity with proximal vs distal subiculum and CA1 (48) .”

*There are several other instances where the chosen formulation obscures a clear line of thinking. The authors clearly think along similar lines when they write in the first paragraph of the Discussion: “These results reveal the functional topography of the human EC as a gateway between neocortex and hippocampus and show remarkable accordance with principles known from anatomical studies of rodents (lateral vs. medial; for reviews see (*[51]; [44]*) and studies of non-human primates (anterior vs. posterior; see e.g.*
[42]*;*
[48]*)*.*”*

We have revised this paragraph in the Discussion:

“These results reveal the functional topography of the human EC as a gateway between neocortex and hippocampus and show remarkable accordance with principles known from anatomical studies of rodents (rostrolateral vs caudomedial; for reviews see (51; 44) and studies of nonhuman primates (anterolateral vs posteromedial; see e.g. [42]; [48]).”

*Unfortunately in referring to the data of, for example Reagh and Yassa, matters become complicated again (third paragraph of the Discussion), since there they copy the nomenclature LEC and MEC from the authors which unfortunately is based on using LEC for a lateral part of EC and MEC for a medial part of EC. In the next sentence the authors than claim that their data demonstrate “that lateral and medial EC exhibit differential functional connectivity with PRC and PHC.” Here they refrain from using either LEC/MEC or their own adopted AL-EC and PM-EC*.

*We strongly suggest that the authors start by defining areas as AL-EC and PM-EC and not to use any indications of LEC and MEC unless clearly defined for each instance*.

We fully agree that using the terms LEC and MEC for lateral and medial sections of the human EC is misleading in this context. We have changed that paragraph and now only utilize the LEC/MEC nomenclature when referring to the rodent anatomy.

“Two recent fMRI studies reported evidence for task-related activation differences between lateral and medial sections of EC in humans […] Our data help to explain these findings by empirically demonstrating that al-EC and pm-EC exhibit differential functional connectivity with PRC and PHC.”

Other major concerns:

*1) The analyses are rigorous and the results are strong. In order to make these data maximally useful for other researchers, it would be nice to include more detail about the location of the border between anterior-medial EC and posterior-lateral EC. For example, some textual description regarding landmarks that* can *be used to identify the border would be useful. Although the data (particularly the data shown in*
Figure 2*), certainly support the existence of a medial-lateral distinction within EC, the accompanying movie shows mostly an anterior-posterior distinction. This potential inconsistency should be clarified*.

We noticed that the lateral-medial dissociation was not obvious in the movie. We therefore changed the movie and added a part in the middle where we provide a top view on the subregions where the medial-lateral distinction is more obvious (Figure 6).

Author response image 1.**DOI:**
http://dx.doi.org/10.7554/eLife.06426.017

In addition, we now also provide al-EC and pm-EC masks, which clearly show the lateral-medial dissociation, for online download. These were created by repeating the multivariate classification approach, which reveals predictions for PRC and PHC preferential connectivity for each voxel, across all subjects. We further describe approximate anatomical boundaries of these al-EC and pm-EC subdivisions. We also provide these EC subregions as masks in template and MNI space. We have added a subsection entitled “Landmarks for delineation of al-EC and pm-EC” in the Results section, which describes the creation of al- and pm-EC masks and their anatomical boundaries.

*2) A similar AP distinction was identified in a single neuron recording study by Killian et al. (Nature 2012)*.

We have added this reference in the Introduction.

3) Given the observations that highlight differential gradients rather than distinctly separate EC areas, is it appropriate to title the paper as revealing “functional subdivisions” of EC?

We do not see a principle contradiction between “subdivisions” and “gradients”. Although gradients define transitions between connectivity profiles in our data, we nevertheless identify regions with differential connectivity (al-EC and pm-EC) which, in our view, is compatible with the term “subdivisions”.